# Mitochondrial Cytochrome *bc*_1_ Complex as Validated Drug Target: A Structural Perspective

**DOI:** 10.3390/tropicalmed9020039

**Published:** 2024-02-01

**Authors:** Lothar Esser, Di Xia

**Affiliations:** Laboratory of Cell Biology, Center for Cancer Research, National Cancer Institute, National Institutes of Health, 37 Convent Drive, Room 2122C, Bethesda, MD 20892, USA

**Keywords:** complex III, cytochrome *bc*_1_ complex, inhibitor binding, cytotoxicity, selectivity

## Abstract

Mitochondrial respiratory chain Complex III, also known as cytochrome *bc*_1_ complex or cyt *bc*_1_, is a validated target not only for antibiotics but also for pesticides and anti-parasitic drugs. Although significant progress has been made in understanding the mechanisms of cyt *bc*_1_ function and inhibition by using various natural and synthetic compounds, important issues remain in overcoming drug resistance in agriculture and in evading cytotoxicity in medicine. In this review, we look at these issues from a structural perspective. After a brief description of the essential and common structural features, we point out the differences among various cyt *bc*_1_ complexes of different organisms, whose structures have been determined to atomic resolution. We use a few examples of cyt *bc*_1_ structures determined via bound inhibitors to illustrate both conformational changes observed and implications to the Q-cycle mechanism of cyt *bc*_1_ function. These structures not only offer views of atomic interactions between cyt *bc*_1_ complexes and inhibitors, but they also provide explanations for drug resistance when structural details are coupled to sequence changes. Examples are provided for exploiting structural differences in evolutionarily conserved enzymes to develop antifungal drugs for selectivity enhancement, which offer a unique perspective on differential interactions that can be exploited to overcome cytotoxicity in treating human infections.

## 1. Introduction

The way we view the function of the mitochondrion and understand its regulation has changed significantly in the past few decades, from its simplistic conception as a cellular powerhouse beginning in the 1960s to the recognition of its contribution to a multitude of cellular biosynthetic and signaling pathways; the latter has been fueled in part by the advancement in structural studies of mitochondrial proteins and by the realization of potential applications via interference in mitochondrial function. The literature regarding structural studies of mitochondrial proteins alone is enormous; therefore, we find it is necessary to focus our discussion on a single mitochondrial protein: the respiratory chain Complex III, otherwise known as the cytochrome *bc*_1_ complex. There are several excellent reviews on various aspects of this essential enzyme, including those of mitochondrial supercomplex organization, the function of the electron transport chain, the structures of complex III, and the mechanisms of the enzyme and its inhibition [1,2,3,4,5]. In this review, we will delve into the structural aspect of using Complex III as a drug target and discuss the important issues that may help overcome drug resistance and cellular toxicity.

### 1.1. The Essentiality of Respiratory Chain Components in the Mitochondrial Function of Eukaryotic Organisms

The mitochondrion fulfills critical functions for its eukaryotic host, not the least of which is the production of copious amounts of ATP by oxidative phosphorylation [1]. While energy conversion is the best-known occupation of this organelle, its protein complement, which numbers up to 1500 (in humans [6]), is known to contribute to essential processes such as amino acid metabolism, pyruvate decarboxylation [7], calcium homeostasis [8], iron incorporation [9] into proteins, sterol and lipid syntheses [10], as well as to support or be part of the stress response, immunity [11], and apoptosis and aging processes [12].

The bacterial origin of the mitochondrion is generally understood to be the product of the fusion of an α-proteobacterium with its archaeal or (pre-)eukaryotic host [13]. In the course of co-evolution, more and more genes were transferred from the mitochondrion to the nucleus. On average, only two percent [14] of the mitochondrial proteins are still encoded by mitochondrial DNA or mtDNA. But while differences between eukaryotic lineages exist, trends and patterns emerge. Plants and fungi still express water-soluble proteins; however, animals have exclusively narrowed expression down to hydrophobic, i.e., membrane-bound proteins [15]. Of the 14 proteins (including the 24-peptide humanin [16]) that are expressed in human mitochondria, 13 are part of the respiratory chain (7 of Complex I, 1 of Complex III, 3 of Complex IV, and 2 of Complex V). In contrast, yeast has lost all genes for Complex I but still maintains 1 for Complex III, 3 for Complex IV and 3 for Complex V. An extreme level of minimalism is seen in the 6 kbp mitochondrial genome of *Plasmodium* [17,18,19] of the phylum *Apicomplexa*. Its mitochondrion expresses only three proteins: 1 of Complex III and 2 of Complex IV. The fact that the last protein-encoding genes resistant to transfer to the nucleus are all found in the respiratory chain places considerable importance on their functions.

### 1.2. The Electron Transport Chain (ETC)

The four well-known proteins of the mitochondrial respiratory chain, or ETC (Figure 1), namely NADH ubiquinone (UQ) oxido-reductase (Complex I), succinate UQ oxido-reductase (Complex II), ubiquinol (UQH_2_) cytochrome *c* (cyt *c*) oxido-reductase (Complex III, or cyt *bc*_1_), and cyt *c* oxidase (Complex IV) are recognized for maintaining the cross mitochondrial inner membrane (MIM) potential (Complexes I, III and IV pump protons) used in ATP synthesis. They are noted for directly connecting to the TCA cycle (Complex I and II), for reducing molecular dioxygen (Complex IV), and for their involvement in the generation of cellular reactive oxygen species (ROS) [5], as well as for their assembly into supercomplexes [20]. Depending on the specific organism, the composition of the ETC may vary, lacking certain complexes but containing additional proteins in support of other metabolic and bypass processes. For example, organisms in the phylum *Apicomplexa* lack type-1 or traditional Complex I proteins but still possess enzymes that are dependent on an oxidized quinone pool, including the monotopic electron transfer flavo protein dihydroorotate dehydrogenase (DHODH) and glycerol-3-phosphate dehydrogenase (G3PDH). Under the loss of proton pumping capability, Complex I may be replaced by a type-2 NADH UQ oxidoreductase (NDH2) in fungi, plants, and algal mitochondria [15].

### 1.3. Mitochondrial Cytochrome bc_1_ or Complex III

In the following account we will be looking at a specific protein of the mitochondrial ETC, namely the cyt *bc*_1_ complex, as it is a proven and valuable target of pesticides and anti-parasitic drugs. The importance of this target is that it has not been invalidated by the existence of alternate oxidase (AOX) in many, but not all, eukaryotic cells [21]. The alternate oxidase may be needed in systems experiencing high levels of stress, but it comes at the cost of reduced efficiency, as the direct oxidation of ubiquinol by molecular dioxygen proceeds here without proton translocation. Complex III oxidizes UQH_2_ to UQ, concomitantly reduces cyt *c*, and, by virtue of employing an energy conserving mechanism, pumps two protons into the IMS per reduced cyt *c*. Mitochondrial cyt *bc*_1_ consists of up to 11 different subunits arranged around a membrane-embedded cyt *b* dimer obeying two-fold symmetry. As a comparison, cyt *bc*_1_ from the photosynthetic bacterium *Rhodobacter sphaeroides* is made up of 4 subunits (Figure 2). All cyt *bc*_1_ complexes carry out the same reaction, which is best described by the Q-cycle mechanism [22] (Figure 3). In short, cyt *bc*_1_ oxidizes and reduces membrane-soluble UQH_2_ and UQ, respectively. The substrates are bound at either the UQH_2_ oxidation site (Q_P_/Q_O_, positive side of the membrane) or at the UQ reduction site (Q_N_/Q_I_, negative side of the membrane). The oxidation of UQH_2_ is aided by the iron-sulfur-protein (ISP) subunit and the cyt *c*_1_ subunit—both of which are essential subunits. The 2:1 ratio of substrate oxidation to reduction is the hallmark of an ingenious mechanism that releases four protons into the IMS while taking up two protons from the matrix side. As Complex III operates near the thermodynamic equilibrium (~200 mV of membrane potential), it is a highly efficient enzyme that maintains an oxidized UQ pool as well as the potential across the membrane.

### 1.4. Issues to Be Addressed Using Complex III as a Drug Target

Because of the essential function of ETC in cells, the use of Complex III (or other components of the ETC) as a drug target is justified [23]. However, issues that limit the application of drugs targeting cyt *bc*_1_ can arise and need to be resolved. One such issue is the increasing appearance of microbial strains that are resistant to Complex III-targeting drugs [23,24,25,26], which render these drugs ineffective. A second issue concerns the cytotoxicity of agents targeting cyt bc1 to mammalian hosts due to structural and functional conservation between enzymes from diverse organisms, limiting their use in treating human and animal infections. Despite the perceived difficulty, drugs that target the cyt *bc*_1_ complex of pathogenic microorganisms in clinical applications do exist, suggesting that these drugs are capable of selectively targeting cyt *bc*_1_ complexes of pathogens over that of the host. In this review, we will examine these issues by summarizing research in the literature that has led to our current understanding of the mechanisms of resistance and selectivity of these drugs from a structural point of view. Differences are revealed in the structures of the cyt *bc*_1_ complexes of a variety of organisms, exposing potential vulnerabilities to pharmacological exploitation. In addition, examples are provided in which structural differences in evolutionarily conserved enzymes were exploited to develop antifungal drugs.

## 2. Natural Compounds Targeting Mitochondrial cyt *bc*_1_ Complex

### 2.1. Complex III as a Target for Natural Compounds

The question of whether mitochondrial proteins are useful drug targets [27] can be analyzed in different ways. Nature, however, has answered this question in the clear affirmative and has targeted such proteins as means to gain competitive advantages and protection. For example, rotenone, a compound produced by plants specifically to fend off leaf-eating caterpillars, is a potent NADH dehydrogenase or Complex I inhibitor [28]. Stigmatellin, from myxobacterium *Stigmatella aurantica*, inhibits Complex I [29] as well, but it is also an effective Complex III inhibitor [30] in the nM concentration range. Even fungi secrete effective Complex III inhibitors in plain view of their own mitochondrial Complex III: the basidiomycete *Strobilurus tencacellus* produces strobilurin [31], which served as the prototype in the development of dozens of commercially successful fungicides [32,33]. Similarly, myxothiazol isolated from myxobacterium *Myxococcus fulvus* was shown to target Complex III [34]. These are specific Q_P_ site inhibitors (Figure 3), but nature has also deployed effective Q_N_ site inhibitors like antimycin [35,36]. In general, that proteins of the ETC are valuable targets can be seen in the 2023 FRAC list [37], which enumerates commercial inhibitors of members of the ETC: 3 for Complex I, 24 for Complex II, and 26 for Complex, III in addition to 3 for the ATP synthase.

But before cyt *bc*_1_ inhibitors were used for pest control and medicines, the discovery of numerous structurally diverse compounds provided valuable initial tools for researchers to characterize ETC components. Many Q_P_ and Q_N_ site inhibitors were used to unravel the complex function of cyt *bc*_1_ [38,39]. That cyt *b* housed two distinct but electronically linked quinol and quinone binding sites was ascertained spectroscopically. In addition, the perplexing oxidant-induced reduction of cyt *b* hemes was understood when antimycin was bound to the Q_N_ site, leading to the proposal of a bifurcated electron transfer (ET) pathway at the Q_P_ site [40,41]. However, the mechanics of the bifurcated ET were not revealed, and possibly could not have been revealed, until the crystal structures of cyt *bc*_1_ complex, and its complex with various inhibitors, were determined [42,43,44,45].

### 2.2. Classification of cyt bc_1_ Inhibitors

Over approximately the past 30 years, a considerable number of cyt *bc*_1_ inhibitors have been discovered or synthesized for use as pesticides or anti-parasitic medications. Cyt *bc*_1_ is easily druggable and, with the continued development of new classes of compounds, the future of such inhibitors is looking bright. It is perhaps not surprising that natural inhibitors like stigmatellin, strobilurin, and antimycin show remarkable similarity to ubiquinol/ubiquinone. Generally, cyt *bc*_1_ inhibitors fall into one of the following chemical categories: for Q_P_ site inhibitors, (1) stigmatellin and related hydroxy naphthoquinones, (2) methoxy acrylates, and (3) oxazolidine-2,4-diones. For Q_N_ site inhibitors, (4) antimycin, (5) pyridines [46], and (6) quinolones [47,48] (Figure 4). As this field is still rapidly developing, additions or changes may have to be made. While so far rare [49], an increasing number of inhibitors (notably quinolones) may eventually inhibit both sites.

It is perhaps more convenient to classify cyt *bc*_1_ inhibitors based on their interaction sites in the cyt *bc*_1_ complex, as well as by the structural changes these inhibitors introduce to the complex [50]. Inhibitors that target the Q_P_ site are called P-site inhibitors and those that bind to the Q_N_ site are called N-site inhibitors. Some inhibitors target both the P- and N-sites, and thus are called PN-site inhibitors. Stigmatellin and myxothiazol are classic P-site inhibitors, whereas antimycin A is a typical N-site inhibitor. NQNO, or 2-n-nonyl-4-hydroxyquinoline N-oxide, was shown to bind to both P- and N-sites and is a PN-site inhibitor. Structural studies of many P-site inhibitors have revealed dramatic conformational changes of the ISP extrinsic domain (ISP-ED) induced by the binding of these inhibitors [50,51]. One type of P-site inhibitor, which includes stigmatellin, UHDBT, and famoxadone, is called a P_f_-type inhibitor because the binding of these inhibitors leads to either immobilization or fixation of ISP-ED. A second type of P-site inhibitor is called a P_m_-type inhibitor because the binding of these inhibitors induces mobility in ISP-ED. Myxothiazol, azoxystrobin, and MOAS (3-methoxy-2-(2-styryl-phenyl)-acrylic acid methyl ester) belong to this group.

**Figure 4 tropicalmed-09-00039-f004:**
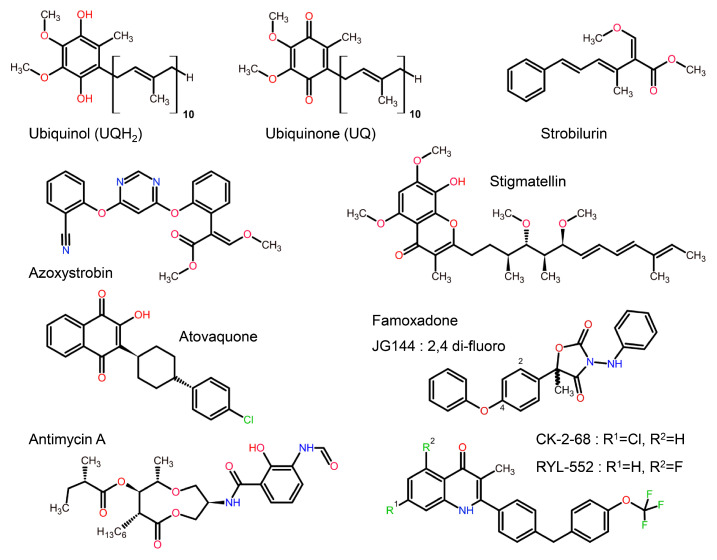
Chemical structures of cyt *bc*_1_ substrate and of a selection of *bc*_1_ inhibitors compared to the substrates ubiquinol and ubiquinone. Compounds that serve as bc_1_ inhibitors may resemble natural substrates (e.g., stigmatellin and atovaquone). In contrast, the natural inhibitor strobilurin and the synthetic famoxadone exploit alternative features of the substrate binding site successfully. Quinolones like CK-2-68 belong to a new class of compounds that extends to endochin-like [52] antimalarials. In this figure, carbon atoms are shown in black, oxygen in red, nitrogen in blue and halide atoms (F or Cl) are shown in green.

### 2.3. Implication of ISP-ED Conformational Transition to the Q-Cycle Mechanism

The Q-cycle mechanism defines two reaction sites in the cyt *bc*_1_ complex [22]: one is the UQH_2_ oxidation or Q_P_ site, and the other is the UQ reduction or Q_N_ site. The most important feature of the Q-cycle mechanism is the bifurcated ET of UQH_2_ at the Q_P_ site, where the two electrons of a substrate UQH_2_ molecule are proposed to diverge in two different paths. The first electron transfers to substrate cyt *c* via the high-potential chain consisting of the ISP subunit (mid-point potential E_m_ = ~250 mV) and cyt *c*_1_ subunit (E_m_ = ~230 mV), whereas the second electron travels through the low-potential heme *b*_L_ (E_m_ = −30 mV) and *b*_H_ (E_m_ = 100 mV) to reach the Q_N_ site, reducing either UQ or UQ^•^. Even though the Q-cycle mechanism best explains such experimental observations, numerous models have been proposed to describe the electron bifurcation at the Q_P_ site [53,54,55,56,57]. The observation of the “inhibitor binding induced conformation switch” of ISP-ED offers a simple mechanism that allows ISP-ED to direct electrons in two different directions [58].

## 3. Pesticide Development

Although many agricultural fungicides are available for disease control, two such fungicides have been particularly well documented in the literature in terms of their development: azoxystrobin and famoxadone, both categorically classified as Qo site inhibitors (QoIs). We use these two fungicides as examples to delineate studies using these compounds and to illustrate their very different modes of action.

### 3.1. Synthetic Fungicides

The development of fungicides based on the natural compound strobilurin A started in 1977 [32,33] but it took until 2004 before detailed structural information (crystal structure) on cyt *bc*_1_ in complex with azoxystrobin became available [50,59]. Following the 1992 introduction of azoxystrobin, which was used as a fungicide to control a broad spectrum of plant pathogens, the great potential of pesticides with an (*E*)-β methoxy acrylate (MOA) toxophore was recognized; this led to more than 10 commercial products, as well as close to 100 derivatives that are currently studied for their utility as fungicides [60].

As an alternative to using naturally occurring compounds as inspiration, knowledge of the fungicidal activity of some heterocyclic compounds, followed by chemical screenings and optimizations combined with greenhouse trials [61,62], led to a new line of cyt *bc*_1_ Q_P_ site inhibitors that compete with and complement MOA-type pesticides. Famoxadone was introduced as the first member of a class of oxazolidine-dione inhibitors. Biochemical and structural characterizations of cyt *bc*_1_ inhibited by famoxadone indicated that this inhibitor used a mechanism different from that employed by azoxystrobin [51,63].

### 3.2. Structural Characterization of the Mechanisms of Inhibition

The crystal structures of bovine mitochondrial cyt *bc*_1_ with bound inhibitors azoxystrobin [50,64] and famoxadone [51,65], respectively, provided the rationale for the resistant mutations found in plant pathogen field strains that were exposed to these fungicides. It is instructive to compare the ways these two inhibitors bind to the Q_P_ site to understand common and distinct features.

There are two separate Q_P_ sites in a dimeric cyt *bc*_1_ complex (Figure 2). The initial structural definition of the Q_P_ site was provided by the structure of cyt *bc*_1_ with bound myxothiazol, where the inhibitor-binding environment consists of the heme *b*_L_, ISP head domain and various conserved residues, many of which are aromatic [42]. A whole host of structural studies of cyt *bc*_1_ inhibitor complexes revealed that each Q_P_ site can be accessed via an opening on an access portal. While the amino acid makeup of the active site is highly conserved, residues lining the surface (Figure 5) of the portal are much less conserved [63,66]. The parts of the Q_P_ site that contain the heme *b*_L_ and that which are close to the access portal are called the proximal and distal portions, respectively.

Both azoxystrobin and famoxadone have a molecular scaffold consisting of three aromatic rings (Figure 4) and bind deeply within the proximal portion of the Q_P_ site, forming a hydrogen bond to the amide N atom of Glu271. Also, in common is the intercalation of an aromatic ring into the narrow gap formed by the highly conserved residues Gly142 and Pro270 at the proximal portion of the Q_P_ site (Figure 6A,B). The dependence on a stabilizing passageway led to the evolution of QoI-resistant fungal strains with a G142A (in yeast sequence G143A) mutation [67]. Finally, as a portion of the inhibitor extends to the entrance portal with its increased sequence variability, significant diversity in the inhibitors’ poses is observed. As illustrated (Figure 6C), the side chain of Phe274 swings around to accommodate the central aromatic rings.

The binding of azoxystrobin and famoxadone, however, induces very different responses from the ISP subunit. A structural study of azoxystrobin binding to cyt *bc*_1_ demonstrated that it induced movement of ISP-ED; thus, it is classified as a P_m_-type inhibitor. By contrast, the binding of famoxadone caused fixation of the ISP-ED, which led to famoxadone being grouped with P_f_-type inhibitors [50,51,58,59,63].

### 3.3. Understanding the Mechanism of Resistance

The utility of structural analyses of this type does not lie in replacing the hard-working chemist with a computer algorithm that would determine the ultimate, never-failing inhibitor. Instead, by knowing how and where an inhibitor binds, we can understand in some detail the occurrence of resistance mutations, why cross-resistance may be a factor, and why some inhibitors have a longer effective lifespan. In our chosen example, one recognized immediately in both types of inhibitors that the hydrogen bond to the backbone amide is likely very advantageous. Protein backbone structure is hard to change through mutation, especially when a change needs to be made at a residue of extraordinary importance in catalysis. Here, the highly conserved residue Glu271 is a general base abstracting one proton from the substrate ubiquinol. The cyt *b* subunit of Complex III is one of the very few remaining proteins encoded in mtDNA and the sole arena for the simultaneous oxidation and reduction of co-enzyme Q substrates (redox pair UQH_2_/UQ). The mitochondrial DNA is known to mutate at a rate ten times higher than nuclear DNA [68]. Thus, rapidly occurring point mutations in the sequence of cyt *b* led to pesticide-resistant strains of plant pathogens.

While the entire environment of the inhibitor binding site can effectively undergo mutation, a few residues stand out as being targeted for mutation with remarkable consistency between different species. This may, however, reflect the limited choice of classes of compounds targeting cyt *bc*_1_ rather than an intrinsic mechanism of response to foreign substrates in cyt *b*. The passageway (Figure 6A,B) between the otherwise highly conserved residues Gly142 and Pro270 often experiences a glycine to alanine mutation. The slightly larger methyl side chain [69] is enough to hinder/prevent the binding of most of the commercial QoIs like azoxystrobin [50,69]. However, the constriction does not abolish ubiquinol oxidation. The mutant strains experience a performance penalty, as there is a drive to reverse back to Gly142 when the exposure to these inhibitors is lifted. Interestingly, a few species even display alanine in their natural sequence (e.g., sea urchins [70]). Recently, G142A-desensitized, loosely MOA-related QoIs with tetrazole-based [71] toxophores have been developed. Still, opposing Gly142 is proline Pro270, which is not known to be mutated as it is part of the PEWY cyt *b*-signature sequence with virtually 100% conservation. Further mutations with the purpose of preventing the inhibitor from binding are slightly more specific to the chemical/physical nature of the pesticide. Mutations with the highest frequency in yeast pathogens can be rationalized by effects of steric hindrance, exchanging beneficial aromatic (π-stacking) interactions for aliphatic ones and in a few cases, aromatic residues are swapped for polar or even charged residues as reviewed here [23,72].

### 3.4. Strategy to Overcome Complex III Resistance

#### 3.4.1. Developing New Classes of Complex III Inhibitors

To overcome drug resistance, several strategies have been deployed. From a structural point of view, both Q_P_ and Q_N_ sites can be targeted. New chemical classes of inhibitors targeting the Q_P_ site are being pursued. These efforts have often been channeled toward specific diseases. For example, medicinal chemistry has focused on *cyt bc_1_* inhibitors of the malarial parasite, which include acridinediones, pyridones, and quinolone aryl esters [73]. Similarly, selective inhibitors targeting the cyt *bc*_1_ complex of pathogenic fungal strains infecting human hosts identified indazoles that have the additional effects of restricting pathogenic strains’ carbon source utilization, macrophage escape, and virulence [74].

Traditionally, the quinone reduction site (Q_N_) is less frequently targeted for disease control; thus, it may be a worthwhile site to target via novel inhibitors. The quinone reduction site (Figure 7) is far more guided by classic (static, lock-and-key) principles compared to the dynamic Q_P_ site that features electron bifurcation. For example, Antimycin A is a high-affinity and specific Q_N_ site inhibitor but lacks selectivity for different organisms. However, new pyridine [46] and quinolone-based [47] inhibitors were discovered to be prospective anti-malarials. Other Q_N_ site inhibitors include the natural antibiotic ilicicolin H from the fungal plant pathogen *Cylindrocladium iliciola* and funiculosin from *Penicillium funiculosum Thom* [75,76]. Interestingly, these compounds exhibited different IC_50_ values depending on the target organisms, suggesting different modes of interaction.

Efforts have also been made to develop compounds that are able to target both Q_P_ and Q_N_ sites One example is ELQ-400 [77,78]. The newly developed ELQ (endochine-like quinolone [78,79,80]) compounds showed promising results in inhibiting cyt *bc*_1_ of mutant strains where classic Q_P_ site inhibitors failed. It is particularly interesting that by manipulating chlorine (Cl) substitutions on the quinolone ring of ELQ compounds, selective targeting of the Q_P_ or Q_N_ site can be achieved for *Pfbc*_1_ [81]. Indeed, structural characterizations of CK-2-68 and CK-2-67, two ELQ-type compounds with the difference of a single Cl substitution on the quinolone ring, bind to Q_P_ and Q_N_ sites, respectively, of *Btbc*_1_ [82,83]. Dual site inhibition has been confirmed only in rare cases, as in ascochlorin [49] and NQNO [84], where structural analyses unequivocally demonstrated binding of these inhibitors to both sites. Genetic or biochemical definitions of site selectivity depend on the rise of escape mutations, which do not exclude binding to the alternate site. Therefore, while the true site can be considered the primary binding site with higher binding affinity, the other site with lower affinity is the secondary site. One obvious advantage of having a higher affinity primary site and a lower affinity secondary site is that the targeted parasite is less likely to generate simultaneous escape mutations at both sites. Thus, dual site inhibitors would provide an excellent way to circumvent or slow down resistance.

#### 3.4.2. Targeting Alternate ETC Components

As research into new cyt *b* inhibitors could be foiled by the complications mentioned above, other enzymes of the respiratory chain have been considered. In the following paragraphs, we will use the multidrug resistant *Plasmodium falciparum* malarial parasite as an example to discuss this approach.

*Plasmodium* mitochondrial ETC replaces Complex I with NADH dehydrogenase type II (*Pf*NDH2) localized in the intermembrane space. But, by virtue of being outside the inner membrane, it does not contribute to the mitochondrial proton motive force. As it reduces the substrate UQ, it is vulnerable to inhibitors mimicking the shape of the natural substrate. The mitochondrial enzyme *Pf*NDH2 was thus considered a potential target [85] in the search for novel drugs as Complex III Q_o_ inhibitor-resistant *Plasmodium* strains emerged. However, the *Pf*NDH2 knock-out parasite was shown to be viable, indicating this enzyme is functionally dispensable [86,87].

More evidence disputing the use of *Pf*NDH2 as an alternate target followed. Although the quinolinone compounds RYL-552 (target confirmation by high-resolution crystal structure [88]) and CK-2-68 [89] are effective and proven inhibitors of *Pf*NDH2 in vitro, *Plasmodium* strains challenged by the new compounds developed resistance mutations in *Pf*cyt *b* [90] but not in the sequence of the intended target *Pf*NDH2. This implies that the new compounds bind in vivo to *Pfbc*_1_. That cyt *b* is indeed the target of CK-2-68 has been confirmed recently by cryoEM [82]. These unexpected “off-target” resistance mutations underline experimental results demonstrating that in the life cycle of the malaria parasite, *Pf*NDH2 is non-essential compared to the *Pfbc*_1_ complex [87,91].

*Plasmodium falciparum* dihydroorotate dehydrogenase (*Pf*DHODH) is involved in pyrimidine biosynthesis, an essential process for the parasite’s survival. It is widely accepted that unlike mammalian cells, the blood-stage malaria parasite relies mainly on glycolytic energy metabolism and so depends on the de novo pyrimidine biosynthesis that is essential for the formation of DNA, RNA, glycoproteins, and phospholipids. Unlike cyt *bc*_1_ inhibitors that tend to resemble ubiquinol/ubiquinone, *Pf*DHODH inhibitors exhibit more diverse scaffolding. The most effective of these is 5-fluoroorotate and its derivatives; however, the combination of 5-fluoroorotate with atovaquone has proven to be more efficient than either compound alone, improving potency and decreasing drug resistance frequency [73].

#### 3.4.3. Combinatorial/Alternate Use of Inhibitors

It should be noted that to minimize the chance for pathogens to develop drug resistance, it is a common practice to use a combinatorial approach to treat patients in clinics (for example, the combination of atovaquone and proguanil [92] remains highly effective) and to spray plants in agriculture for microbial infections. This approach has been demonstrated in the treatment of HIV patients, in the protection against malarial parasites [73], and in the wide-spread application of fungicides in crop protection [93].

## 4. Target cyt *bc*_1_ of Human and Animal Pathogens

Because of their high functional and structural conservation, cyt *bc*_1_ complexes are largely considered unsuitable targets when it comes to treating microbial infections in humans or animals. One governing principle in developing anti-parasite/fungal agents has been the avoidance of those targets shared by pathogens and hosts due the compounds’ known [94] or foreseeable cytotoxicity. Exceptions do exist, such as the FDA-approved drug atovaquone that specifically targets Complex III of human pathogens and the commonly used veterinary medicine decoquinate, which also targets Complex III of the coccidia parasite in treating animal infections [95,96]. It is therefore highly beneficial, both in terms of scientific curiosity and medical practicality, to study the underlying principle of the selectivity afforded by this drug. Such studies could also lead to new directions for future drug development, especially those targeting Complex III for treatment of fungal and parasite infections in humans and animals.

### 4.1. Medicinal Compounds Targeting Mitochondrial cyt bc_1_ of Human, Malaria, and Fungal Pathogens

Within the list of the 848 currently-known human parasites [97], *Plasmodium falciparum* stands out, as it causes nearly half a million deaths per year. Carried by the mosquito of the genus anopheles are the malaria-causing protozoans *P. falciparum*, *P. malariae*, *P. vivax*, *P. ovale,* and *P. knowlesi*. *Plasmodium*, as an obligate parasite, has a complex life cycle and depends on blood-feeding mosquitoes and vertebrate hosts [98]. Once it enters a human host, significant parasitic growth occurs in the blood cells or erythrocyte stage, during which it can be treated with various types of medication. Atovaquone (Figure 4) is a cyt *bc*_1_ inhibitor (Figure 8) effective in the treatment and prevention of malaria, including chloroquine-resistant strains [99]. The parasite contains a single mitochondrion in the erythrocyte stage and possibly a few in the schizont stage. In the blood stage of the parasite, experiments have shown that the ETC is not used or required for ATP production [100]. Rather it is needed for maintaining an oxidized quinone pool for the use of dihydroorotate dehydrogenase (DHODH) in the de novo pyrimidine synthesis—a required function for the parasite as it lacks the metabolic pathways to salvage pyrimidine. Nevertheless, the function of mitochondrial cyt *bc*_1_ remains essential and thus it is a valuable drug target.

Interestingly, atovaquone was also approved for use in treating fungal infections in humans, including pneumonia caused by *Pneumocystis jiroveci* (PCP), which is the type of pneumonia most likely to affect teenagers and adults with human immunodeficiency virus (HIV). Fungal infections are common in patients with compromised immune systems due to viral infections, organ transplantations, kidney failure, etc. Incidents of invasive fungal diseases such as candidiasis and cryptococcosis are on the rise; this trend is expected to continue because the number of patients at risk for fungal infections is increasing as immunomodulatory therapies continue to expand and our ability to support highly immunocompromised patients improves. Unfortunately, treatment options for fungal infections are limited. As an example, the standard therapy for cryptococcosis, one of the most prevalent, invasive, and life-threatening fungal infections on the planet, is currently based on drugs such as amphotericin that were developed in the 1950s. Since then, only two additional classes of antifungals have been developed. This rate of antifungal drug discovery is unlikely to meet future demands.

### 4.2. Structural Studies of Selective cyt bc_1_ Inhibitors

Selectivity of drugs for cyt *bc*_1_ of *Plasmodium falciparum* (*Pfbc*_1_) in the presence of high concentrations of host cyt *bc*_1_ is a significantly greater challenge than in crop protection, where only minimal amounts of pesticides remain at the time of product consumption. Unfortunately, the structures of *Pfbc*_1_ alone and its complex with atovaquone are not available, perhaps due to challenges in overexpression and purification of *Pfbc*_1_ in sufficient quantities and quality from cultured pathogenic *plasmodium* strains. However, it was shown that yeast (*Saccharomyces cerevisiae*) cyt *bc*_1_ (*Scbc*_1_) has a sequence similar to *Pfbc*_1_ and its atovaquone binding could be studied via crystallography and spectroscopy [66]. The dramatic Y268S/N/C mutation [101] in *Pfbc*_1_ that results in >6700-fold resistance to atovaquone may be explained by a significant loss of interaction with the hydroxynaphtho quinone moiety of atovaquone. Extensive cyt *b* sequence alignments demonstrated a function-driven, exceedingly high conservation [66] of residues in the active site but relaxed conservation at the Q_P_ access portal.

Even more dramatic are the structural studies of the fungicide famoxadone. Both *Bos taurus* mitochondrial cyt *bc*_1_ (*Btbc*_1_) and *Rhodobacter sphaeroides* (*Rsbc*_1_) can be inhibited by famoxadone. But unlike classic QoIs such as stigmatellin and myxothiazol that inhibit *Btbc*_1_ and *Rsbc*_1_ equally well, famoxadone showed a significant difference in inhibiting *Btbc*_1_ and *Rsbc*_1_, favoring the latter with an IC_50_ ratio of >300 [63]. Crystal structures of *Btbc*_1_ and *Rsbc*_1_ with bound famoxadone revealed that this difference in binding derived from the terminal phenoxy group of famoxadone, which remains at the access portals and interacts differently with residues in these two enzymes.

### 4.3. Use of Bacterial and Mitochondrial cyt bc_1_ to Search for Selective QoIs

While the efficacy of QoIs is important, we need to direct our attention to specificity. Approved pesticides for crop protection are meant to target a broad spectrum of fungal disease agents, but they may pose a risk to humans and animals. Studies have shown that azoxystrobin and famoxadone inhibit *Btbc*_1_, which is highly homologous to human Complex III, but allows structure determinations. In research, however, isolated and purified fungal Complex III is not always readily available for experimentation. This prompted researchers to see if *Rsbc*_1_ (Figure 2) could stand in for fungal mitochondrial Complex III. Sequence analyses have shown better agreements between cyt *b* of *Rsbc*_1_ and *Scbc*_1_ than *Btbc*_1_. It was found that famoxadone inhibits *Btbc*_1_ at IC_50_ = 418 nM but inhibits *Rsbc*_1_ at the much lower concentration of IC_50_ = 1.4 nM. Given the considerable overall sequence conservation in cyt *b*, it becomes important to understand the origin of a nearly 300-fold inhibition ratio [63]. Two residues near the Q_P_ portal essentially swap their character, as shown in Figure 9. The *Bt* cyt *b* residue Phe 276 destabilizes famoxadone by not being close enough to lend support. The situation reverses as Phe 301 of *Rs* cyt *b* sterically interferes unless the tail group (Figure 9) flips positions as observed. Analysis was aided by the observation that the famoxadone-related compounds jg144 and fenamidone inhibit both enzymes in the low nM range and feature inhibition ratios (*Btbc*_1_/*Rsbc*_1_) of 1 and 2.5, respectively [63]. As the entire ubiquinol oxidation site features only highly or very highly conserved residues (Figure 5A), inhibitors that are meant to be species specific must be large enough to interact with residues at the access portal to the Q_P_ site, where strict conservation ceases. Thus, the small jg144 and fenamidone compounds inhibit *Btbc*_1_ and *Rsbc*_1_ equally well, whereas famoxadone is just large enough to reach into a variable region at the entrance of the Q_P_ site (Figure 5B).

## 5. Challenges in Drug Design and Future Developments

The mitochondrion has maintained a degree of independence; more importantly, it has proven to be indispensable. Decades of research have established the essentiality of the mitochondrial ETC in cellular function, qualifying components of the ETC as targets for small molecule intervention. Despite the fact that ETC inhibitors have been used widely in research and in control of pathogenic diseases in agriculture, the challenges in overcoming resistance and their use in medicine remain formidable. Mechanisms of drug resistance have been well studied in the treatment of cancer and pathogenic microbial infections and are characterized by the cell’s versatility in dealing with chemical stress. However, options for overcoming drug resistance remain limited. While the approach in current practice is to screen pathogens for an ever-expanding set of inhibitors, such an approach requires further lengthy and inefficient identifications of drug targets. Structure-based drug design, which takes advantage of the vast knowledge of the structural and conformational space of Complex III, seems to be advantageous in obtaining specific lead compounds more efficiently. Inhibitors targeting the ETC apart from Complex III for medical applications are even more limited. In this paper, we have outlined an approach that seems useful, at least in principle, in overcoming the problem of specificity by taking advantage of the structural diversity in regions of Complex III that are less conserved between pathogens and humans. Also, on the forefront of structural biology, cryoEM techniques are poised to enable the determination of much sought-after pathogenic enzymes, even if only minute amounts are available. This is an exciting and promising area of active research for compounds that have enhanced selectivity.

## Figures and Tables

**Figure 1 tropicalmed-09-00039-f001:**
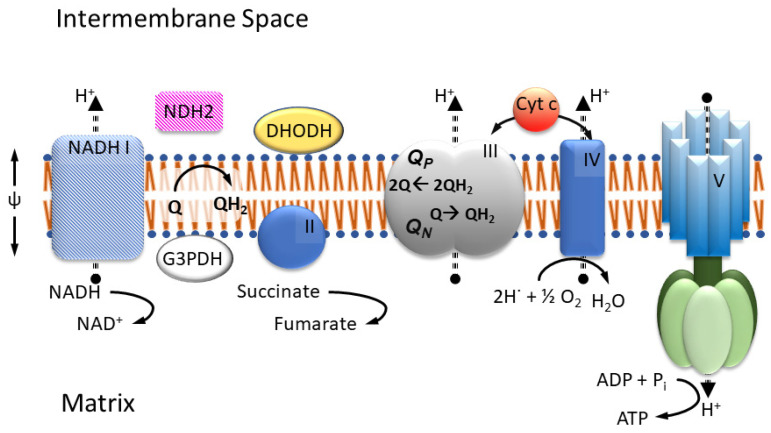
Mitochondrial respiratory chain. NADH-dehydrogenase (I), Succinate-dehydrogenase (II), Cytochrome *bc*_1_ (III), and Cytochrome c oxidase (IV) are embedded in the inner membrane together with ATP synthase (V). In most fungi and parasites, complex I is replaced by type II NADH dehydrogenase. As enzymes beyond complex I and II (e.g., dihydro orotate dehydrogenase (DHODH) or glyceraldehyde 3-phosphate dehydrogenase (G3PDH) in some organisms) depend on the availability of oxidized ubiquinone, the proper function of complex III, with its two respective sites for the oxidation Q_P_ and reduction Q_N_ of quinol/quinone, is critically important.

**Figure 2 tropicalmed-09-00039-f002:**
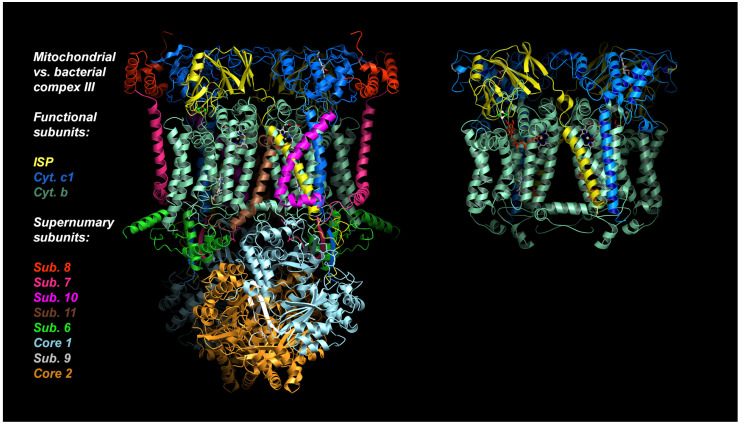
Structures of mitochondrial and bacterial cyt *bc*_1_**.** Structure of bovine mitochondrial cyt *bc*_1_ with 11 subunits, compared to its bacterial ancestor *Rhodobacter Sphaeroides* cyt *bc*_1_ with 4 subunits (3 shown). The essential core is made up of just 3 proteins: Cytochrome *b*, cytochrome *c*_1,_ and Rieske iron-sulfur-protein (ISP).

**Figure 3 tropicalmed-09-00039-f003:**
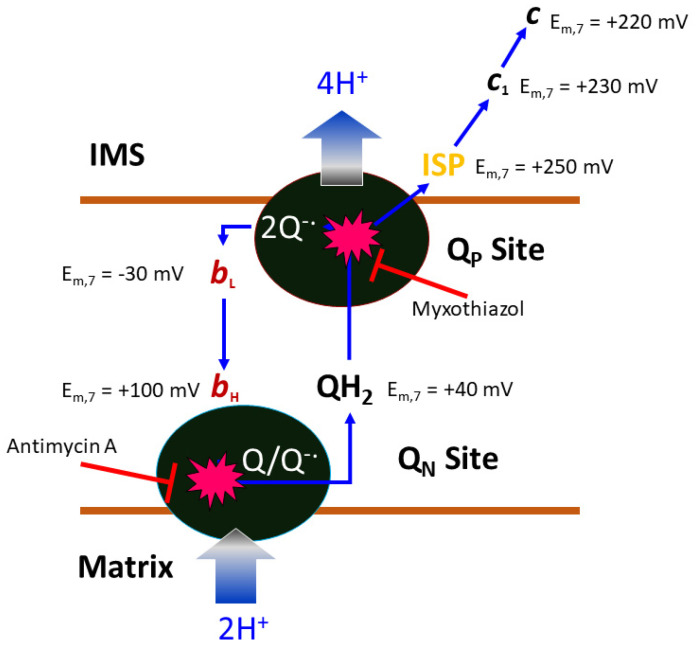
Illustration of the Q-cycle mechanism. Within each half of the two-fold symmetric complex of bovine mitochondrial cyt *bc*_1_, there are three essential subunits: cyt *b*, cyt *c*_1_, and ISP. Mid-point potentials for the *c*_1_ heme (230 mV) and 2Fe2S cluster of ISP (250 mV) are given. The potentials for substrate cyt *c* (*c*, 220 mV) and ubiquinone (QH_2_, 40 mV) are given. The membrane-embedded cyt *b* subunit contains two active sites with access to opposite sides of the bilayer (MIM). At the intermembrane space side (positive side), the quinol oxidation site (Q_P_ site) oxidizes ubiquinol to ubiquinone, while at the matrix side (negative side), the quinone reduction site (Q_N_ site), ubiquinone is reduced to ubiquinol. In a completed Q-cycle, the number of quinol molecules oxidized and quinone molecules reduced is 2:1. The Q_P_ site carries a low-potential heme group (*b*_L_, −30 mV), which is electronically coupled to the high-potential heme (*b*_H_, 100 mV) group of the Q_N_ site. Critical to the Q-cycle mechanism is the bifurcated electron transfer of the substrate QH_2_ at the Q_P_ site, in which the first electron follows the high-potential chain (ISP, *c*_1_, and *c*) and the second electron travels to the low-potential chain (*b*_L_ and *b*_H_). Both Q_P_ and Q_N_ sites can be inhibited by competitive inhibitors like myxothiazol and antimycin, respectively. This cycle accounts for the uptake of two protons from the matrix, the flow of two electrons from the Q_P_ to the Q_N_ site, and the release of four protons to the IMS, contributing significantly to the membrane potential.

**Figure 5 tropicalmed-09-00039-f005:**
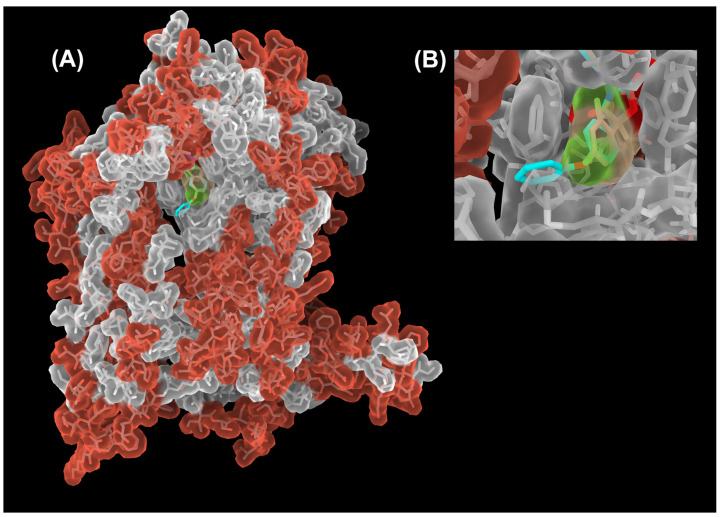
Sequence conservation of the cyt *b* subunit between *Btbc*_1_ and *Rsbc*_1_. (**A**) Structure of the cyt *b* subunit is rendered as a semi-transparent molecular surface overlaid on its stick model. Residues that are conserved between *Btbc*_1_ and *Rsbc*_1_ are shown in gray. Those that are different are colored red. The core of the cyt *b* subunit (light gray surface) is well conserved overall, especially in the Q_P_ site. A decline in conservation starts outside the Q_P_ site in the access portal, to which the terminal phenoxy group (cyan) of the bound famoxadone (stick model) is exposed. (**B**) Close-up view of the portal (slightly rotated up for a better view of the inside), showing that a superposition of the small jg144 inhibitor (stick model in yellow) is unable to reach the outside of the Q_P_ pocket and is therefore unable to differentiate between *Btbc*_1_ and *Rsbc*_1_. This is different for famoxadone with its phenoxy-phenyl moiety (shown as stick model in cyan) that is just long enough to detect non-conserved residues outside of the portal.

**Figure 6 tropicalmed-09-00039-f006:**
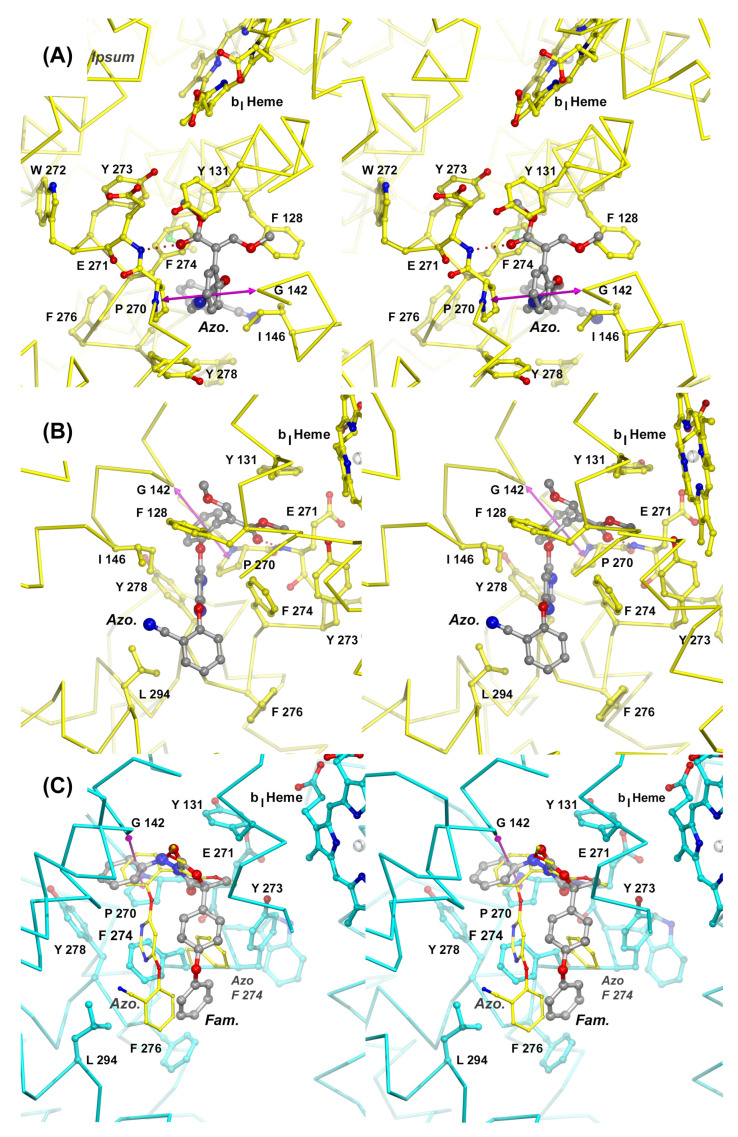
Mechanisms of inhibition of Complex III by azoxystrobin and famoxadone. Stereographic diagrams of the Q_P_ site show the structure of the cyt *b* subunit as a Cα trace centered on the inhibitor binding pocket. Panels (**A**) and (**B**) provide two views of the Q_P_ site with bound azoxystrobin (Azo). The Cα trace of subunit *b* is colored yellow. The inhibitor is rendered as a stick model in gray and is shown hydrogen-bonded to the backbone of Glu271 as indicated by a dotted line. Residues that are participating in interactions with the inhibitor are shown as stick models with carbon atoms in yellow, oxygen in red, and nitrogen in blue. The magenta double arrow spans the distance between the center of Pro270 and Gly142. A simple Gly to Ala mutation (G142A, G143A fungal seq.) is responsible for the rise of QoI-resistant fungal strains. Panel (**C**) shows the superposition of two Q_P_ site inhibitors, azoxystrobin (Azo) and famoxadone (Fam). While there is significant spatial overlap in the most confined portion of the active site, the large Q_P_ site rearranges (F274), allowing the inhibitors to diverge.

**Figure 7 tropicalmed-09-00039-f007:**
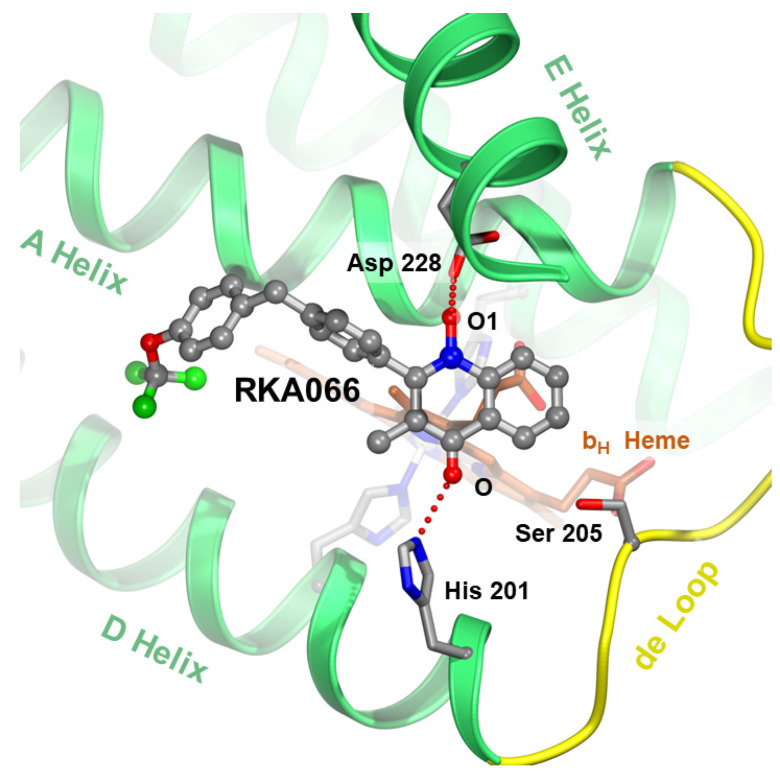
Targeting the ubiquinone reduction (Q_N_) site. The new quinolone compound RKA066 [48] at the Q_N_ site is shown as a stick model with carbon atoms in gray, oxygen in red, nitrogen in blue, and fluorine in green. The competitive inhibitor hydrogen-bonds with critical residues D228 and H201 (in stick models) and brings the electron flow to the b_H_ heme (brown stick model) to a halt. In addition, the newly developed hydroxy quinolinone/ol inhibitors are suspected to bind to both Q_N_ (as shown) and Q_P_ (awaits verification) sites.

**Figure 8 tropicalmed-09-00039-f008:**
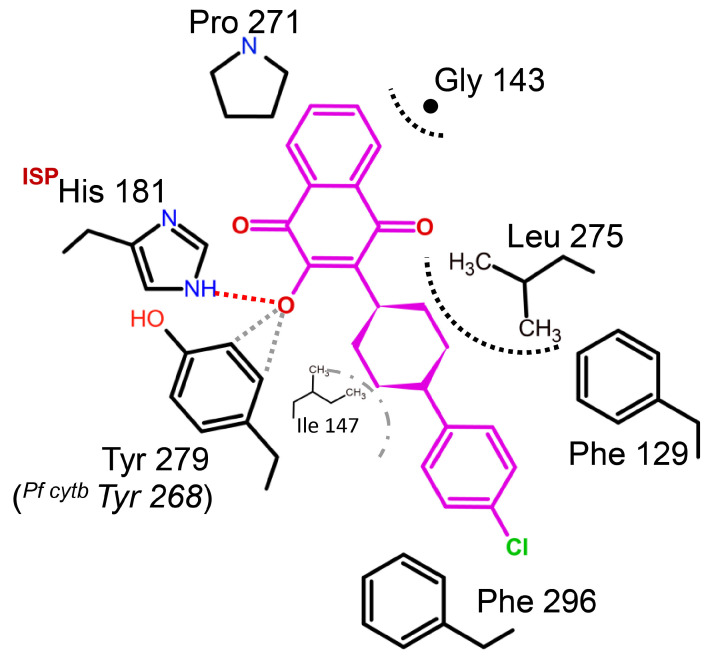
Binding environment of atovaquone in *Saccharomyces cerevisiae*. The crystal structure analysis of *Scbc*_1_:atovaquone shows that the inhibitor binds at the Q_P_ site. Atovaquone is a highly effective medication used to treat malaria, but fails dramatically to inhibit ^PF^Y268S (or C, N) mutant strains. Tyrosine residue plays an important role in stabilizing the deprotonated hydroxynaphtho-quinone. Its structure suggests possible aromatic C-H to hydroxyl interactions (gray dotted lines) might explain atovaquone’s sensitivity to site mutations.

**Figure 9 tropicalmed-09-00039-f009:**
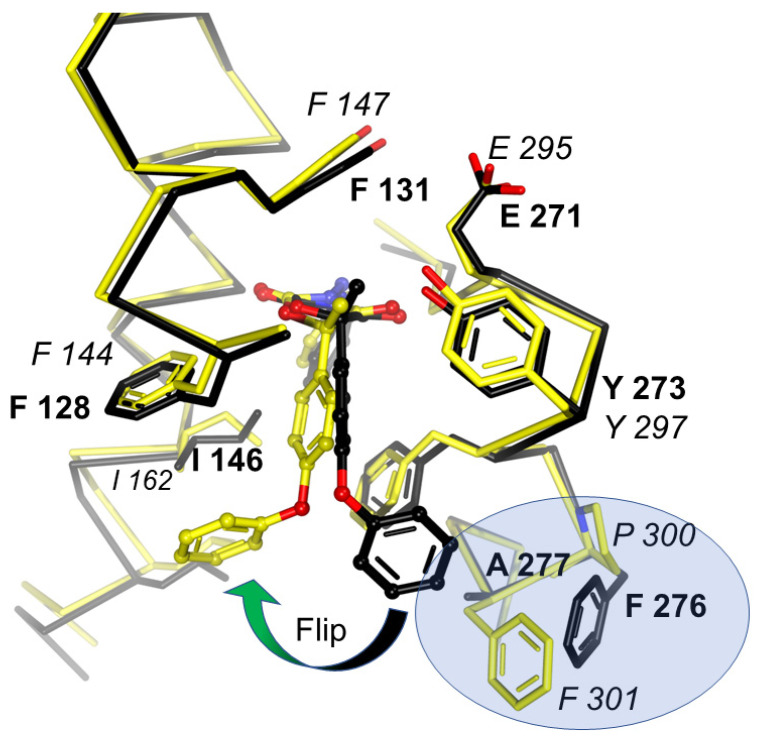
Comparison of famoxadone binding to *Btbc*_1_ and *Rsbc*_1_**.** Binding of famoxadone to *Btbc*_1_ (black model) and *Rsbc*_1_ (yellow model) shown with superposed Cα traces. While the proximal pocket of Q_P_ is constructed by highly conserved residues, greater variability of residues at the entrance is responsible for the differential binding and inhibition. Residues that determine the pose of famoxadone are FA in *Btbc*_1_ and PF in *Rsbc*_1_, which are highlighted in the blue oval.

## Data Availability

No new data were created or analyzed in this study. Data sharing is not applicable to this article.

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
