# Peer review of "Mitochondrial Cytochrome bc1 Complex as Validated Drug Target: A Structural Perspective"

_tropicalmed, 2024, doi:10.3390/tropicalmed9020039_

Round 1
Reviewer 1 Report
Comments and Suggestions for Authors
The paper summarises key inhibitors
developed towards the mitochondrial ETC
bc1 complex, further discussing
challenges associated with toxicity,
resistance, and approaches for future

Reviewer 2 Report
Comments and Suggestions for Authors
The review is well written, but I have a few suggestions.
- Consider improving the numbering and descriptions of the figures. For instance, provide the title of each figure before its description. Instead of 'Fig 6. B shows,' ensure that after the figure has an appropriate identifier/description, then individual sub figs e.g. A, B, C, and D are then described/titled.
- The color of the gray dotted line in Figure 8 appears to be relatively light. Consider using a darker shade for better visibility and contrast.
- References 63 and 65 are missing and incomplete.
Additional Comments:
The main thrust of the review is on the structural features of the mitochondrial respiratory chain Complex III, a validated target for fungicides and antibiotics. The review noting advancements in its mechanisms and inhibition, addresses issues related to cytotoxicity, mechanism of resistance development, and drug selectivity via structural comparisons of inhibitor and target binding in different organisms. The work sheds light on drug resistance and offers ideas on Q-cycle mechanisms by looking at examples of structures with bound inhibitors. In addition, it proposes how structural variations can help create antifungal medications, opening a new approach to overcoming cytotoxicity in the treatment of infections in humans. Using examples found in the protozoan parasite Plasmodium, the reviewers address some of the issues raised.
Although the authors mention that a few reviews have been written about the topic. They provide a more concise review from a structural perspective, which is unique.
The references are appropriate. Although the authors have been informed about missing references.
Some of the figures need corrections. I had asked the authors to consider improving the numbering and descriptions of some of the figures.
Reviewer 3 Report
Comments and Suggestions for Authors
This is a great subject for a review. For compounds that target cyt bc1, it is important to relate observed selectivity / resistance generation to sequence and structure, especially where CryoEM and crystal structures are available. The authors did this well for azoxystrobin and famoxadone, but they did not do so for the other compounds and organisms in this review. There is plenty of selectivity and structure data out there for other compounds (ATV, RYLs, CKs, ELQs, etc), including for resistant mutants and host / parasite enzymes. Not having treated all of the compounds in the same way that azoxystrobin and famoxadone were treated is a missed opportunity.
Overall, the writing is very good but, unfortunately, the review is poorly organized, the content is not explained well by the figures and some of the figures are either not discussed or are discussed at the wrong place.
Specific concerns follow:
L194 Numbering of inhibitor types is wrong. Should be… “(4) antimycins, (5) pyridones”.
L195 Category (6) is more correctly ‘quinolones’, not ‘quinolinones’.
L212 Figure 4. It should be made clear that the ‘Tail – Linker/Spacer – Toxophore’ model applies to azoxystrobin and famoxadone only. In this context, it is not necessary to show the structure of azoxystrobin twice, and the authors should take this opportunity to define the famoxadone toxophore. Need to show structures of native ligands, UQ and UQH2. Need to discuss how structures of known inhibitors relate to structures of UQ / UQH2.
L254 What is the famoxadone ‘toxophore’ referred to here? No such toxophore was described in the famoxadone section.
L258-266 Need a figure that shows what is being described here.
L268 – What is the famoxadone ‘toxophore’?
L289 Figure 6 is very large. Consider breaking into two, especially as it already has two separate figure captions. Frames 6B, 6C and 6D are not referenced in the text of the review until the very last section. That section should be moved here.
L350 It is strange to omit the most well-known cyt bc1 Qi-site targeting quinolone, MMV167, which is in the MMV’s drug development pipeline.
L355-362 This analysis is completely incorrect. Stickles et al. described ELQ-300 resistant Pf D1 with a mutation at the Qi-site of Pf cyt bc1. They suggested a Qi/Qo-site targeting combination therapy (atovaquone and ELQ-300) to overcome both ATV and ELQ-300 resistance. Perhaps this combination therapy discussion belongs in section 3.4.3.
L364 Figure 7. RKA066 was not mentioned in the text of the review and yet it occurs in this figure.
L378 No, Pf NDH2 is not currently considered a valid Pf drug target for reasons given in lines 380-388.
L375-388 These paragraphs should not be included in a section titled ‘targeting alternate ETC components’, because they are about compounds that have been shown to target cyt bc1. In fact, I suggest removing this section, because the authors are specifically focusing on cyt bc1 inhibition.
L394 This is the first time that the authors mention that cyt bc1 inhibitors tend to resemble UQ and UQH2. This needs to be dealt with much earlier.
L408 If the authors are going to bring up animals, then they should probably also mention decoquinate, which is commonly used to treat coccidiosis infections in animals.
L418-422 The authors have discussed Pf a number of times before, and yet they wait until now to talk about malaria and the plasmodium life cycle. This discussion needs to occur earlier.
L448 Figure 8. The structure of atovaquone is wrong. The ring attached directly to the hydroxynaphthoquinone should be completely saturated (not aromatic).
L449 Why is the Y268S mutation first mentioned here? It should be discussed in the Qi/Qo-site selectivity section.
L416 Should there be a comma between ‘human’ and ‘malaria’?
Also, in the section about host / parasite selectivity, why is atovaquone’s host activity not discussed? The question the authors should be dealing with is… why is ATV non-toxic to humans even though it has a sub-micromolar IC50 vs the mammalian enzyme? This is a hugely important question within the context of this section. How important is host/parasite selectivity when the only approved drug shows relatively high inhibition of the host enzyme, or is there another factor that makes ATV safe (efflux, etc.)?
Also, other cyt bc1 active compounds (such as MMV167) have been shown to have high selectivity for the parasite vs. host enzyme. What was the structural basis of this observed selectivity? It should be discussed here, especially as the sequence difference that accounts for this selectivity also explains the relatively poor activity of MMV167 vs. toxoplasmosis.
Also, given the importance of host / parasite selectivity, why do more publications not include this data? Should authors who publish papers about new compounds that target cyt bc1 be encouraged to focus more on potential host toxicity?
L454 It is strange to have a separate section entitled ‘structural studies of selective cyt bc1 inhibitors’, when this is sort of the point of the whole review. Why not fold these observations into previous sections, or why not include some of the material from other sections here? For example, the authors mentioned that there are Pf mutants that are resistant to the CK and RYL compounds and there are structural studies that show that these compounds bind to the Qo-site. Why are these structural studies of selective cyt bc1 inhibitors not included here?
L458-460 Several labs have published Pf cyt bc1 enzyme activity data using protein isolated from Pf. Thus, though it may not yet be possible to isolate enough material to do CryoEM, it certainly is possible to isolate enough to do enzyme selectivity studies.
L462-464 The Y268S mutation is not ‘medically devastating’ as it was circumvented by ATV / proguanil combination therapy.
L478 In a review article, it is odd to see a sentence begin with, “As our studies have shown…” The authors seem to have shifted into a first person research narrative in this section, which is unusual to see in a review context.
L490-498 Why do the authors have their discussion of Figure 6A-D here rather than earlier in the review where the discussion of Figure 6 actually occurred. There is nothing about their selectivity discussion that could not have been brought up earlier. In fact, given that this review is supposed to be a ‘structural perspective’, shouldn’t structure-based selectivity analysis be a core part of the paper for all of the compounds, not something that’s brought up just at the end?
Comments on the Quality of English LanguageThe writing was very good. Minimal English language editing necessary.
Round 2
Reviewer 3 Report
Comments and Suggestions for Authors
I'm fine with the edited version.